# Expression Levels of Heat-Shock Proteins in *Apis mellifera jemenetica* and *Apis mellifera carnica* Foragers in the Desert Climate of Saudi Arabia

**DOI:** 10.3390/insects14050432

**Published:** 2023-04-30

**Authors:** Ahmad A. Alghamdi, Yehya Z. Alattal

**Affiliations:** Department of Plant Protection, Chair of Engineer Abdullah Ahmad Bagshan for Bee Research, College of Food and Agriculture Sciences, King Saud University, Riyadh 11587, Saudi Arabia

**Keywords:** heat-shock protein, thermo-tolerance, qPCR, *hsp70*, *hsp28*, *hsc70*, *hsp90*, *hsp10*

## Abstract

**Simple Summary:**

*A. m. jemenetica* occurs naturally on the Arabian Peninsula and in tropical Africa, and this honeybee subspecies has acquired several morphological, behavioral and molecular adaptations to extreme summer temperatures in Saudi Arabia. In this study, expression levels of different heat-shock protein (*hsp*) genes in forager *A. m. jemenetica* (a thermotolerant honeybee subspecies) and *A. m. carnica* (a thermosusceptible subspecies) were explored and compared under desert and semi-arid climates within Saudi Arabia. The results revealed higher expression levels of *hsp* mRNAs in *A. m. jemenetica* compared to *A. m. carnica*. The expression levels of small- as well as large-molecular-weight heat-shock proteins were higher under desert climate conditions in the Riyadh region compared to the semi-arid conditions in Baha. It is clear that the expression of heat-shock proteins is a key molecular mechanism of *A. m. jemenetica* adaptation to extreme summer conditions.

**Abstract:**

*A. m. jemenetica* is the indigenous honeybee of the Arabian Peninsula. It is highly adapted to extreme temperatures exceeding 40 °C, yet important molecular aspects of its adaptation are not well documented. In this study we quantify relative expression levels of small- and large-molecular-weight heat-shock proteins (*hsp10*, *hsp28*, *hsp70*, *hsp83*, *hsp90* and *hsc70* (mRNAs)) in the thermos-tolerant *A. m. jemenetica* and thermosusceptible *A. m. carnica* forager honeybee subspecies under desert (Riyadh) and semi-arid (Baha) summer conditions. The results showed significant day-long higher expression levels of *hsp* mRNAs in *A. m. jemenetica* compared to *A. m. carnica* under the same conditions. In Baha, the expression levels were very modest in both subspecies compared those in Riyadh though the expression levels were higher *in A. m. jemenetica*. The results also revealed a significant interaction between subspecies, which indicated milder stress conditions in Baha. In conclusion, the higher expression levels of *hsp10*, *hsp28*, *hsp70ab*, *hsp83* and *hsp90* mRNAs in *A. m. jemenetica* are key elements in the adaptive nature of *A. m. jemenetica* to local conditions that enhance its survival and fitness in high summer temperatures.

## 1. Introduction

Scientists do not know exactly how global warming will influence beekeeping, but it is likely to be an additional stress factor [1,2] that may necessitate unique beekeeping practices and alter the natural distribution of the honeybee subspecies (*A. mellifera*) worldwide. According to the National Oceanic and Atmospheric Administration (NOAA) [3], uneven warming across the Earth will cause more regions to experience rising rather than cooling temperature trends. In regions such as Saudi Arabia, a significant increase in the maximum average daily temperature (0.71 °C/decade) has been reported [4], with the average summer temperature exceeding 40 °C [5]. Furthermore, the general drought conditions that dominate most of the Arabian Peninsula may exacerbate the influence of increasing temperatures on beekeeping and beekeeping practices in this region.

The common honeybee *Apis mellifera* occurs naturally in Asia, Europe and Africa and has spread worldwide because of modern beekeeping practices [6,7]. Thirty geographical honeybee subspecies of *Apis mellifera* have been identified in diverse climatic zones and thermal gradients [6,8]. These subspecies display different thermal responses to extreme temperatures that are associated with adaptation to warmer climates. This leads to better fitness of these populations at higher temperatures relative to populations from temperate areas [9]. Recently, an Asian origin of *A. mellifera* was supported via an adaptive radiation involving selection on a few genomic “hotspots” [10]. These findings emphasized the importance of Middle Eastern honeybee subspecies such as *A. m. syriaca* and *A. m. jemenetica* in the evolution and radiation of *A. mellifera* [10].

Because the *Apis mellifera* honeybee colony has a eusocial structure, it exhibits unique responses to extreme temperatures, such as fanning [11,12,13,14], clustering [6,14], reduced foraging, stinging or even migratory swarming [6]. Thermal homeostasis of the honeybee colony is also essential for brood development because *A. mellifera* larvae and pupae are extremely stenothermic [15,16]. In addition to eusocial responses, thermotolerant honeybees have acquired specific morphological and molecular modifications that enhance their ability to withstand extreme temperatures. Body size and heat-shock protein synthesis are two of the documented adaptive traits responsible for heat tolerance in worker bees [6,10,17]. These traits can significantly diminish the impact of extreme temperatures on thermotolerant honey bee subspecies along with the unique social behavior of *A. mellifera* colony [18,19]. Therefore, *A. mellifera* is viewed as a super-organism and a model for functional homoeothermic insects [11,16,20,21].

*A. m. jemenetica,* an indigenous honeybee subspecies to the Arabian Peninsula and tropical Africa [22,23,24], is the only honeybee subspecies that occurs naturally in Asia and Africa. On the Arabian Peninsula, it is believed that *A. m. jemenetica* beekeeping goes back about three thousand years [25]. It exhibits very distinctive morphological and behavioral features compared with other honeybee subspecies. Its comparatively small body [6], higher capability to forage at extreme temperatures [26] and ability to survive drought conditions with minimal food storage [25] are some of its unique characteristics [6]. Consequently, many scholars have characterized *A. m. jemenetica* as a highly thermotolerant subspecies [6,27,28,29]; specifically, its populations in Saudi Arabia and Sudan have been documented as being the most heat tolerant among *A. mellifera* subspecies [6]. However, its thermotolerance might diminish because of intensified climate change in Saudi Arabia [4,30]. Moreover, more than 1.3 million exotic honeybee packages of other subspecies and hybrids are imported annually into Saudi Arabia to fulfil seasonal beekeeping demand [31], which has a negative impact on local population structures [32] and a direct cost exceeding USD 40 million [31]. Most of these imported packages die in the summer months due to extreme temperature and drought [29]. Therefore, determining the heat-resistance mechanisms and tolerance thresholds of different honeybee subspecies would be very useful and may necessitate the conservation of many native *A. mellifera* subspecies and result in the reconsideration of importing exotic honeybee subspecies to regions that have well-established honeybee subspecies and extreme climatic conditions.

The heat-resistance mechanisms of *Apis mellifera* can include genetic or epigenetic modifications of the individual honeybee [20,21,33]. Heat-shock proteins (HSPs) are expressed in the cells of all organisms [34] and play a key role in the thermoregulation of insects, including *Apis mellifera* [35,36,37]. The expression of *hsp* genes can be induced in response to environmental cues such as heat shock, pesticides, oxidants, ultraviolet radiation and biotic stresses [35]. Exposure to sub-lethal heat stress suppresses protein synthesis in the cell and stimulates the transcription of heat-shock protein synthesis, while exposure to a lethally high temperature leads to apoptosis, which HSPs prevent by inducing protein thermal stability [38].

Based on their molecular mass and function, HSPs are classified into six families: HSP20 (small HSPS), HSP40 (J-proteins), HSP60, HSP70, HSP90 and HSP100 [37]. The expression of heat-shock proteins to increased ambient temperatures [39] is typically rapid, but unstressed cells keep the transcription of the inducible heat-shock proteins (HSPs) inactive. In a recent study, the exposure of *A. m. jemnetica* nurse bees to sublethal heat induced the transcriptional activation of many heat-shock protein genes by remodeling histone methylation states [33]. While some heat-shock proteins (HSPs) are induced by environmental stressors, others are continuously expressed during normal cell function [35]. For example, heat-shock cognate 70 (*Hsc70*) is constitutively expressed in the organism’s cells, while *hsp70* is triggered only by stressors [40,41]. Under heat stress situations, HSP70 and all other inducible heat-shock proteins go into fast cap-independent translation pathways to increase the efficiency of heat-shock protein synthesis and to refold the large amount of misfolded proteins [42]. When the temperature returns to normal, the newly synthesized HSPs continue refolding misfolded proteins for some time and then resume regular transcription pathways [42].

The increased expression of *hsp*s may contribute to species-specific stress tolerance and may involve one or more stress proteins [17]. For example, their differential expression in *Apis mellifera*, *A. cerana*, *A. florea* and *A. dorsata* in response to heat stress has been reported [43]. The expression of *hsp*s may increase survival under heat or other environmental stressors and may also influence individual fitness in specific habitats [17,44]. In relation to the common honeybee *A. mellifera*, which has 30 geographical subspecies, the expression of heat-shock proteins could be subspecies-specific and may explain the disparity in heat tolerance and ability to survive extreme temperatures. In a two-year observational study in Saudi Arabia, survival rates among three different honeybee subspecies (*A. m. jemenetica*, *A. m. carnica* and *A. m. ligustica*) were highly associated with temperature range, and most colony losses (92% for *A. m. carnica*, 84% for *A. m. ligustica* and 48% for *A. m. jemenetica*) occurred in the summer [29]. Furthermore, *A. m. jemenetica* showed higher heat response thresholds compared to *A. m. carnica* or *A. m. ligustica* [45,46]. The Carniolan honeybee *A. m. carnica* is native to the temperate European climate, ranging from the Alps to the Carpathian Mountains [6], and it adapts poorly to temperature extremes. Nevertheless, the mechanisms for how *A. m. jemenetica* can withstand extreme summer temperatures are not well clarified. Indeed, Saudi Arabia is ideal for conducting comparative studies among honeybee subspecies and exploring their adaptation to extreme ambient temperatures and drought.

In this study, the quantitative expression levels of *Hsp10*, *Hsp28*, *Hsp70*, *Hsp83* and *Hsp90* in the indigenous thermotolerant *A. m. jemenetica* were measured and compared with expression levels in the thermosusceptible *A. m. carnica* under summer foraging conditions. The study was conducted in two climatic regions—Riyadh (desert) and Baha (semi-arid) (https://en.wikipedia.org/wiki/Climate_of_Saudi_Arabia, accessed on 1 February 2023)—and expression levels were measured at three-day intervals. We hypothesized that the levels of HSP gene expression would be much higher in *A. m. jemenetica* compared to *A. m carnica* under both conditions.

## 2. Materials and Methods

### 2.1. Study Sites

The study was conducted in two thermogeographical regions within Saudi Arabia: Riyadh (latitude: 24.7742 N, longitude: 46.7385 E), which provided a perfect hot, dry desert habitat, and Baha (latitude: 20.0046 N, longitude: 41.283.6 E), which has a semiarid climate (Table 1). Riyadh experiences extremely hot summers with an average maximum temperature of around 43 °C. Baha, a highland area in the Sarawat Mountain Range (called Sarah) at an altitude of 2270 m above sea level, has milder summers and may offer a better habitat for *A. m. carnica* (www.latlong.net, accessed on 1 March 2022).

### 2.2. Honeybee Colonies Preparations

Sixteen colonies of *A. m. jemenetica* were obtained from purebred certified local breeders (chair of bee research, King Saud University). Another sixteen colonies of *A. m. carnica* were established by heading 16 queenless package bees with imported purebred certified Carniolan queens (LOKACIJA, Slovenia). After three months of colony establishment, subspecies affiliation for each colony was confirmed by our lab at the university’s Bee Research Unit by conducting a morphometric analysis of body size, wing angles and the color of the abdominal tergites using 15 honeybees per colony [47,48]. Reference data for both subspecies were obtained from the Oberursel Bee Research Institute (Frankfurt, Germany). Eight colonies of each subspecies were kept in each of the two thermogeographical regions. After standardization, each colony consisted of 3–4 brood frames and 7–8 adult bee frames. Colonies were established in each location in March and were treated according to the International Federation of Beekeepers’ Associations (Apimondia) guidelines for performance testing until sampling [22].

### 2.3. Forager Honeybee Sampling

In June, each of the 32 established colonies (16 *A. m. carnica* and 16 *A. m. jemenetica*) was sampled 3 times. The first samples were collected in the morning one hour after sunrise (about 7:00 a.m.); the second group about midday (12:00 p.m.); and the third group at 5:00 p.m. Samples of 10 foraging bees were taken at the entrance of each honeybee colony using small forceps, dipped directly in liquid nitrogen and stored in a freezer at −80 °C. Then, the samples representing the same region, race and sampling time were mixed together to form a single pooled sample. In total, 12 pooled samples were set, 6 from Riyadh and 6 from Baha, each comprising 3 samples for each subspecies. A pooled sample consisted of 80 bees from 8 colonies to be used for gene expression analysis. Ambient temperatures were recorded in the study locations at sampling times.

### 2.4. RNA Extraction and cDNA Synthesis

Total RNA was extracted from each pooled sample (bee heads and thoraces) using TRIzol™ Plus RNA Purification Kit following the manufacturer’s instructions (Invitrogen, Carlsbad, CA, USA) and was further purified using Qiagen RNeasy column (Qiagen, Germantown, TN, USA). First-strand cDNA was synthesized using the SuperScriptTM III First-Strand Synthesis Super Mix (Invitrogen, Carlsbad, CA, USA) according to the manufacturer’s guidelines. The synthesized cDNA was used as a template in real PCR reactions.

### 2.5. Primer Design and Real-Time PCR

Public databases (NCBI, Honeybee Genome Consortium) were explored for known heat-shock protein (HSP) genes. The genes included were: *Hsp10*, *Hsp28*, *Hsp70*, *Hsp83* and *Hsp90*, and heat-shock cognate-70 *Hsc70cb*. The gene sequences were downloaded into Geneious^®^ Prime v.2019.2.3 (https://www.geneious.com, Biomatters Ltd., Newark, NJ, USA, accessed on 1 January 2020) for analysis and primer design. The designed primers are listed in Table 2. Briefly, real-time PCR tests were performed using SYBR GREEN (SYBR^®^ GREEN PCR Master Mix; API: Applied Biosystems, Carlsbad, CA, USA) and the 7500 Real-Time PCR System (Applied Biosystems, Carlsbad, CA, USA). The real time reaction mix (25 μL) was prepared from 13 μL of the master mix, 2 μL of forward primer (2 pmol), 2 μL of reverse primer (2 pmol), 2 μL of the cDNA of the sample and 7 μL of nuclease-free water. The *A. mellifera β*-actin gene was used as an endogenous control for relative expression data analyses. Reactions were conducted in triplicate. Amplification was performed under the following qPCR thermocycling conditions: 95 °C for 5 min, 40 cycles of 95 °C for 10 s, 30 s at 57 °C, 72 °C for 10 s and 95 °C for 20 s.

### 2.6. Statistical Analysis

Relative expression and fold change analyses were performed using qPCR Ct values for each gene according to the sampling plan. Relevant actin cycle threshold (Ct) values were used to calculate the relative expression (relative expression = 2 ^ct (actin-gene)^). Average ct values of *A. m. carnica* samples from Baha were used as a calibrator to calculate gene expression fold changes in different heat-shock protein genes (fold change = 2 ^ct (reference-(actin-gene)^). Significant fold change differences were determined based on the average gene expression of each group using SAS Statistical Analysis System software suite (SAS Institute: www.sas.com, accessed on 15 February 2023). Mixed models included the random effects of honeybee subspecies, location and sampling times followed by Multiple Student–Newman–Keuls tests. Differences were considered statistically significant at *p* < 0.05, and the figures were prepared using GraphPad Prism 9 (www.graphstates.net, accessed on 5 January 2023).

## 3. Results

Significantly higher expression levels of five *hsp* mRNAs (*hsp70ab*, *hsc70cb*, *hsp83*, *hsp90* and *hsp28*) were found in *A. m. jemenetica* compared to *A. m. carnica* (*p* values for all comparisons are listed in Table 3). The expression levels of *hsp10* mRNA were higher in *A. m. jemenetica* as well, but the variation was not significant (P > F = 0.0520). The highest variation in the *hsp* mRNA fold change occurred in the *Hsp70* mRNAs (more than 150×) using the expression levels of *A. m. carnica* in Baha as a calibrator, which revealed the highest absolute expression levels among all other heat-shock protein genes (Figure 1). Nevertheless, the smaller-molecular-weight *hsp10 and hsp28* exhibited higher expression fold changes compared to the larger-molecular-weight *hsp83 and hsp90* in this study. Fold changes in the relative expression of *hsc70cb mRNAs* were the lowest in both subspecies.

The results also revealed significantly higher relative expression levels of all *hsp* mRNAs under desert conditions (Riyadh) compared to semi-arid conditions (Baha) (Table 2). Within the same region, relative expression levels were higher in *A. m. jemenetica* except in the only case of *hsc70cb* in Baha, which was significantly higher in *A. m. carnica* (Figure 1). Although variations in the relative expression levels between *A. m. jemenetica* and *A. m. carnica* within the same region were significant for both Riyadh and Baha, variations in the relative expression levels between both subspecies were far higher under desert conditions (Figure 1). The relative expression levels of all *hsp* mRNAs (using the expression levels of *A. m. carnica* in Baha as a calibrator) were very modest in Baha mainly for *hsp70ab, hsp10* and *hsp28* (Figure 1). The highest difference in the relative expression levels within the same region was calculated for *hsp70ab*. The results also revealed a significant interaction between subspecies and region in the relative expression levels of *hsp70ab*, *hsp90*, *hsp83* and *hsc70cb* (Table 1).

In Riyadh, the ambient sampling temperatures were 31, 42 and 39 °C (morning, midday and evening, respectively). In Baha, they were 22, 30 and 25 °C. Variation in the expression levels of all *hsp* mRNAs were also significant among sampling times (Table 1). However, relative expression levels of *hsp* mRNAs were not consistent for either subspecies within the same sampling time (Figure 1). The highest relative expression levels were not always at the highest day temperature; some were in the morning then evening in both locations (*hsc70cb* and *hsp28*), while others exhibited the highest expression levels at midday (*hsp10* and *hsp83)* and in the evening (*hsp70ab)* in Riyadh (Figure 1).

## 4. Discussion

The study shows day-long higher expression levels of *hsp* mRNAs in the thermo-tolerant honeybee subspecies *A. m. jemenetica* compared to the thermosusceptible *A. m. carnica* under the same conditions. This indicates a higher adaptation of *A. m. jemenetica* to local conditions at both locations. Under the milder ambient temperatures in Baha, the expression levels were very modest in both honeybee subspecies though the expression levels for *A. m. jemenetica* were still higher. The higher expression levels of *hsp10*, *hsp28*, *hsp70ab*, *hsp83* and *hsp90* mRNAs in *A. m. jemenetica* in Baha can be associated with the adaptation of *A. m. jemenetica* to start heat-shock protein synthesis even at lower foraging temperatures, which enhances survival and fitness at higher temperatures and might be associated with longer foraging trips. The accumulation of *hsp70* mRNAs at a low foraging temperature was reported in the desert ant *Cataglyphis bombycina*, which can forage at an ambient temperature exceeding 50 °C [49]. Furthermore, the real body temperature of foraging *A. m. carnica* and *Melipona panamica* was found to be higher than the reported ambient temperatures [50,51], and this may be the same for *Apis mellifera*. The foragers’ body temperature could also be affected by foraging duration distance and conditions. Nevertheless, the modest expression levels in Baha and the significant interaction between subspecies and location indicated less heat-stress-inducing conditions to *A. m. carnica* compared to Riyadh. Consequently, keeping *A. m. carnica* in Baha can be more successful compared to Riyadh, where extremely low year-round survival rates (8%) were reported in the summer months [29]. Obviously, the higher expression levels of *hsp* mRNAs in *A. m. jemenetica* under desert foraging conditions can be proposed as a key component of higher adaptation to extreme temperatures compared to *A. m. carnica*. Both honeybee subspecies exhibited significantly different expression levels of *hsp* mRNAs at different sampling times, which indicated the fast and continuous adjustment of *hsp* expression in response to daily changes in ambient foraging temperatures.

In this study, the relative expression levels of *hsp70*ab mRNA were the highest among the other *hsp*s. Heat-shock proteins of the HSP70 family demonstrated a very fast response and exhibited robust transcriptional activation and expression in many insects after heat stress [33,42,45]. Transcriptional induction of the HSP70 family was associated with the overall transcriptional downregulation of constitutively expressed genes such as heat-shock cognate 70Cb (Hsc70cb). Under desert conditions, the fold change in *Hsp70* mRNA was more than 50× higher in *A. m. jemenetica* than *A. m. carnica* and might be considered the main heat-shock protein to be involved in thermotolerance. This could also explain the generally lower expression levels of the constitutively expressed *hsc70cb* in both regions. The fold change differences in *Hsc70cb* mRNA were the lowest between both subspecies as well as locations, which could be related to the relatively small transcriptional activation response of *Hsc70cb* as a constitutive stress gene to the increase in ambient foraging temperatures, and its expression is associated with other biological processes [52]. The special cap-independent translation pathway of *hsp70* [42] may explain the high fold change in the relative expression of *hsp70* mRNA under hot foraging in Riyadh compared to the relatively normal foraging temperature in Baha. The relative amounts of *hsp70* mRNA increased during the day along with the increase in the ambient temperature, but they resumed regular transcription pathways at night when all forager bees had returned to the colony and *hsp70* mRNA began to decay. This was also applicable to the inducible small heat-shock proteins (HSPs). In a previous study [45] using the SDS-PAGE method, the expression of the large-molecular-weight heat-shock proteins hsp70 and hsp83 were reported in nurses of *Apis mellifera jemenetica* after exposure to 40 °C, but only hsp70 in foragers after exposure to 45 °C [45]. In this study, samples were collected when forager bees returned to their colonies with body temperatures normally higher than the reported ambient temperature (14-44°C) [51], which may explain the detection of *hsp70* and *hsp83* mRNAs at lower ambient temperatures in both subspecies. This deviation might also be explained by the inherent technical limitations associated with the WB method such as incomplete protein transfers [53]. Fold changes in the relative expression levels of *hsp28* and *hsp10* mRNAs between hot foraging (Riyadh) and normal foraging (Baha) conditions demonstrated the significant contribution of these heat-shock proteins in enabling both subspecies to overcome heat stress.

On the other hand, the inconsistency in the relative expression patterns of *hsp* mRNAs at different sampling times might be associated with a variable quantitative response and the interaction of different *hsp* genes with short-term environmental changes. We do not know if each *hsp* or *hsp* family has the same response threshold for heat stress and if their interaction among different honeybee subspecies is similar.

In conclusion, the higher expression levels of *hsp10*, *hsp28*, *hsp70ab*, *hsp83* and *hsp90* mRNAs in *A. m. jemenetica* can be strongly associated with adaptation to the extreme ambient desert temperatures which characterize the Riyadh region.

## Figures and Tables

**Figure 1 insects-14-00432-f001:**
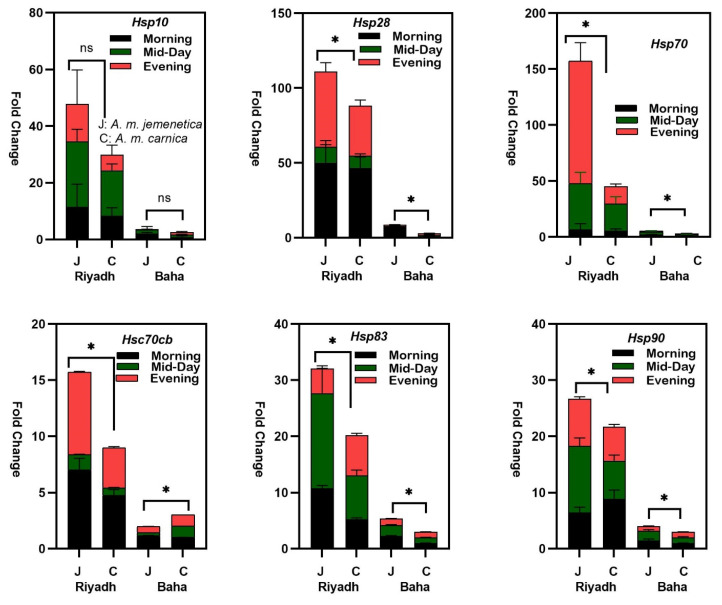
Fold change in *Hsp28*, *Hsp10*, *Hsp83*, *Hsp90*, *Hsp70* and *Hsc70cb* (mRNA) relative expression levels in *A. m. jemenetica* (J) and *A. m. carnica* (C) at the two thermogeographical regions (Riyadh and Baha) and the three foraging times (7:00 a.m.: after ≈ 1 h of foraging; 12:00 p.m.; and 5:00 p.m. (≈1 h before sunset)). Actine was used as endogenous control to calculate ∆ct, and relative expression in *A. m. carnica* in Baha was used as a reference to calculate the fold change (fold change = 2 ^ct (reference-treatment)^). Significant fold change difference was determined based on average gene expression of each group using a three-way ANOVA analysis followed by Multiple Student–Newman–Keuls tests. Differences were considered statistically significant at *p* < 0.05. (*) = significant variation and (ns) = non-significant variation.

**Table 1 insects-14-00432-t001:** Some environmental data from the two study locations, Riyadh and Baha (Source: climate-data.org, accessed on 1 January 2023).

Sampling Position	Altitude (m)	Annual Minimum Temperature (°C)	Annual Maximum Temperature (°C)	Annual Precipitation (mm)	Longitude (E)	Longitude (N)
Riyadh	600	20	44	101	46.71	24.71
Baha	2270	14	26	630	41.63	20.30

**Table 2 insects-14-00432-t002:** Primer sequences designed to amplify different heat-shock proteins with gene names, gene subcellular location and NCBI gene ID. Heat-shock cognate (hsc70cb) genes were used in the present study. (*) subcellular locations of different heat shock genes were obtained from the official website of the University of Port Harcourt: UniPort website (www.uniprot.org, accessed on 1 January 2023).

Locus/Gene Identifier	Cellular Location *	Gene/DNA Region	Primers
LOC724487	Cytoplasm, Nucleus, cytosol	28 KDa heat- and acid-stable phosphoprotein-like	F-GAGGAACCCAAAGCACATGGT R-TCTACACCCTTTGTTTTTCCCTGT
LOC552531	Mitochonderial Matrix	10 KDa heat-shock protein, mitochondrial-like	F-AGCAATTGGACCTGGACAAAGA R-GCCAGTATATCTGACTCACGGAAT
LOC410620	Nucleus, cytoplasm, cytoskeleton	Heat-shock protein 70Ab	F-TGGCATTCCACCTGCACCTA R-TGGTGATCTTGTTCTCCTTTCCAGT
LOC408706	Cytoplasm, mitochondria, nucleus, endoplasmic reticulum	Heat-shock cognate 70Cb ortholog	F-CGCGCGTCTACACGTTCTTT R-CGTGATTTTGATGCCGCAGT
LOC411700	Cytoplasm	Heat-shock protein 83-like	F-TCCACATCTTCTGCTTTTGTTTCC R-TCAACGCGCGTCTTCATTCA
LOC408928	Nucleus, cytoplasm, cell membrane, melanosome	Heat-shock protein 90	F-TGGATCCGTGAGAGATTCATAGCG R-CGCTTTCCAAGCTGAAATTGCACA
NM_0011 85146.1	Cytoplasm, nucleus, cytoskeleton	Apis mellifera actin (Arp1)	F-CGTAAAGATTTGTATGCCAACACTGTC R-AATCCATACGGAATATTTCCTCTCGG

**Table 3 insects-14-00432-t003:** Summary of SAS PROC GLM analyses for different heat-shock proteins (*hsp*s mRNA) relative expression levels. Hypothesis testing was performed using three-way ANOVA (subspecies: *A. m. jemenetica or A. m. carnica*; location: Riyadh or Baha; sampling time: morning, midday or evening. *F*-ratios were calculated from type III mean squares for all statistics. *p* < 0.05 was deemed statistically significant and *p* < 0.001 as highly significant.

Variation Factor	Subspecies		Location		Sampling Time Point		Subspecies by Location	
Heat sock protein	F	P > F	F	P > F	F	P > F	F	P > F
*Hsp*70ab	89.59	0.0001	255.23	0.0001	63.02	0.0001	81.85	0.0001
*Hsc70*cb	77.32	0.0001	919.83	0.0001	235.79	0.0001	142.59	0.0001
*Hsp*83	29.14	0.0001	277.15	0.0001	21.61	0.0001	13.06	0.0014
*Hsp*90	15.54	0.0006	740.73	0.0001	8.62	0.0015	6.58	0.0170
*Hsp*10	4.22	0.0520	60.100	0.0001	5.02	0.0150	3.37	0.0789
*Hsp*28	7.42	0.0119	325.52	0.0001	54.76	0.0001	2.63	0.1179

## Data Availability

Data are available in tables and figures.

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
