# Peer review of "Expression Levels of Heat-Shock Proteins in Apis mellifera jemenetica and Apis mellifera carnica Foragers in the Desert Climate of Saudi Arabia"

_insects, 2023, doi:10.3390/insects14050432_

Round 1

Reviewer 1 Report

Dear Authors,

I find the study carried out with interest to readers of Insects. However, I have some questions.

1. Morphometric analysis of bees.

The authors refer to Meixner et al. [48], which provides methods for identifying subspecies of bees, but no standards for different subspecies.

Authors should provide a link to the article(s) based on which the A.m.carnica standard was used.

Did the authors use the A. m. carnica standard presented in Bouga et al., or other data?

(Bouga M., et al. A review of methods for discrimination of honey bee populations as applied to European beekeeping. Journal of Apicultural Research. 2011. V. 50(1) P. 51-84. doi: 10.3896/IBRA.1.50.1.06).

2. How do the authors explain the differences in Hsp70 expression found in the present study and publication by Alqarni et al.?

Alqarni et al. showed that «The forager bees exhibited differential expression of HSPs after heat stress. No HSPs was expressed in the foragers of A. m. jemenitica, and Hsp70 was expressed only in the foragers of A. m. ligustica and A. m. carnica at 40°C. A prominent diversity in HSPs expression was also exhibited in the foragers at 45°C with one HSP (Hsp70) in A. m. jemenitica, two HSPs (Hsp40 and Hsp70) in A. m. carnica, and three HSPs (Hsp40, Hsp60 and Hsp70) in A. m. ligustica». Thus, Hsp70 is expressed in A. m. jemenitica only at a temperature of 45°C.

The manuscript shows that the collection of bees was carried out in the morning, mid-day, and evening (temperatures at sampling times were 31, 42 and 39⁰C, respectively), that is, at temperatures were below 45°C (line 250-251).

Author Response

Dear Reviewer,

The authors would like to thank you for your comments and suggestions. 

1- question one (Methodology) the source of standard morphometric data (reference data)  of both honeybee subspecies used in confirming affiliation of our samples  was obtained from Oberursel Bee Research Institute (Frankfurt, Germany) via prof. Stefan Fuchs. this part was added in methods and Acknowledgment as well.

2-We believe that the reasons for such deviation between our results and previous results  (Alqarni study) could be related to two points as it was suggested by other researchers: honey body temperature is always higher than ambient temperature by 26-3 degrees,  and to the  inherent technical limitations associated with the WB method such as such as incomplete protein transfer used in Alqarni study.

Reviewer 2 Report

A. m. jemenetica is naturally occurring in the Arabian Peninsula and tropical Africa. In this study, expression levels of different heat shock protein genes (hsp) in forager A. m. jemenetica (thermo-tolerant honeybee subspecies) and A. m. carnica (Thermo-susceptible) were explored and compared under desert and semi-arid climates within Saudi Arabia. Results revealed higher expression levels of hsp mRNAs in A. m. jemenetica compared to A. m. carnica. Expression levels of small as well as large molecular weight heat shock proteins were higher under desert climate conditions encountered by Riyadh region compared to semi-arid conditions in Baha.

Here I suggest some minor revisions.

1.       Forager honeybees sampling, the sampling part is not clear. Is the first group totally different with the second group and the third group? or the colonies for collecting samples are the same colonies at noon collection? Also, the 12 pooled samples are not clear.

2.       Data availability statement, in this section, if there is no data deposit in public database, this section is not necessary.

3.       Acknowledgments, In this section you can acknowledge any support given which is not covered by the author contribution or funding sections. Not to thanks the funding again.

Author Response

Dear Reviewer, 

We are pleased to thank you for your comments and time spend in the revision of our MS. regarding your suggestion to further explain the sampling and how we established the pooled samples, it was added in clearer form. Regarding the paragraphs in the Acknowledgment and data availability, all was corrected.

Best Regards

Yehya

Reviewer 3 Report

I think that in general the work is important and contributes significantly to the knowledge regarding the adaptation of species. I suggest improving the wording of the introduction by trying to summarize the text a bit, making it easier to read. In addition to improving and expanding the discussion. I find continuous bugs about the use of parentheses.

Author Response

Dear Reviewer,

thanks a lot for your comments and time spend in the revision, all corrections in the introduction and discussion parts were done.

Best Regards

Yehya